

# Transformer-based tokenization for IoT traffic classification across diverse network environments

Firdaus Afifi[1,2], Faiz Zaki[2], Hazim Hanif[2,3], Nik Aqil[2] and Nor Badrul Anuar[2,4]

[1] Faculty of Computer Science and Mathematics, Universiti Malaysia Terengganu, Kuala Nerus, Terengganu, Malaysia
[2] Centre of Research for Cyber Security and Network (CSNET), Faculty of Computer Science and Information Technology, Universiti Malaya, Kuala Lumpur, Malaysia
[3] Department of Software Engineering, Faculty of Computer Science and Information Technology, Universiti Malaya, Kuala Lumpur, Malaysia
[4] Institute of Informatics & Computing in Energy, Universiti Tenaga Nasional, Kajang, Selangor, Malaysia

## ABSTRACT

The rapid expansion of the Internet of Things (IoT) has significantly increased the volume and diversity of network traffic, making accurate IoT traffic classification crucial for maintaining network security and efficiency. However, existing traffic classification methods, including traditional machine learning and deep learning approaches, often exhibit critical limitations, such as insufficient generalization across diverse IoT environments, dependency on extensive labelled datasets, and susceptibility to overfitting in dynamic scenarios. While recent transformer-based models show promise in capturing contextual information, they typically rely on standard tokenization, which is ill-suited for the irregular nature of IoT traffic and often remains confined to single-purpose tasks. To address these challenges, this study introduces MIND-IoT, a novel and scalable framework for classifying generalized IoT traffic. MIND-IoT employs a hybrid architecture that combines Transformer-based models for capturing long-range dependencies and convolutional neural networks (CNNs) for efficient local feature extraction. A key innovation is IoT-Tokenize, a custom tokenization pipeline designed to preserve the structural semantics of network flows by converting statistical traffic features into semantically meaningful feature-value pairs. The framework operates in two phases: a pre-training phase utilizing masked language modeling (MLM) on large-scale IoT data (UNSW IoT Traces and MonIoTr) to learn robust representations and a fine-tuning phase that adapts the model to specific classification tasks, including binary IoT *vs.* non-IoT classification, IoT category classification, and device identification. Comprehensive evaluation across multiple diverse datasets (IoT Sentinel, YourThings, and IoT-FCSIT, in addition to the pre-training datasets) demonstrates MIND-IoT's superior performance, robustness, and adaptability compared to traditional methods. The model achieves an accuracy of up to 98.14% and a 97.85% F1-score, demonstrating its ability to classify new datasets and adapt to emerging tasks with minimal fine-tuning and remarkable efficiency. This research positions MIND-IoT as a highly effective and scalable solution for real-world IoT traffic classification challenges.

Corresponding authors
Firdaus Afifi, fdaus.isa@gmail.com
Nor Badrul Anuar,
badrul@um.edu.my

# INTRODUCTION

The rapid growth of the Internet of Things (IoT) ecosystem has substantially transformed the technological landscape, leading to a significant surge in both the volume and diversity of network traffic. Recent data indicates there are approximately 17.08 billion connected IoT devices globally, which is projected to nearly double to 29.42 billion by 2030 (*Vailshery, 2024*). Subsequently, these phenomena make accurate IoT traffic classification essential for maintaining network security, optimizing resource allocation, and ensuring efficient device management (*Tahaei et al., 2020*). IoT traffic classification involves identifying the types of data packets transmitted across IoT networks. Failure to correctly classify or detect anomalous traffic can lead to security breaches, inefficient network usage, and degraded performance, particularly in high-stakes sectors such as healthcare, smart cities, and industrial automation.

Therefore, various approaches in the domain aim to address this growing demand. Early methods relied heavily on port-based and payload-based techniques, but their effectiveness diminished with the rise of encrypted and obfuscated IoT traffic. Subsequently, attention shifted to machine learning approaches, including decision trees, support vector machines, and k-nearest neighbours, which offered incremental improvements but often remained highly dataset-specific, failing to generalize across diverse IoT environments and device types (*Aqil et al., 2022*). More recently, deep learning techniques, particularly convolutional neural networks (CNNs) and recurrent neural networks (RNNs), demonstrated better results due to their ability to capture complex traffic patterns (*Lotfollahi et al., 2020*; *Thapa & Duraipandian, 2021*). Despite these advancements, the dynamic and evolving network characteristics prove to be a significant limitation to existing deep learning solutions. A critical gap remains in their inability to adequately handle generalization across different dynamic IoT networks and device types without extensive retraining or costly data annotation.

The deep learning models' dependency on labeled datasets further worsens this issue, particularly in dynamic IoT environments where new device types frequently emerge and data privacy restrictions are stringent. This highlights the urgent need for models that can efficiently learn and adapt to generalized IoT traffic classification. To overcome these limitations, leveraging pre-training techniques emerges as a promising solution, particularly effective in handling the complexities and heterogeneity of IoT networks by enabling models to learn robust and generalized representations from large-scale data. Within this paradigm, transformer-based models demonstrate significant success in capturing long-range dependencies and contextual information, further enhancing generalization across diverse IoT traffic conditions (*Wang & Li, 2021*; *Wu et al., 2022*). However, their full potential in IoT traffic classification is hindered by their reliance on standard tokenization strategies. Unlike natural language, IoT traffic is characterized by

irregular and heterogeneous data structures, often consisting of numerical features and statistical flow attributes rather than sequential words. Standard tokenization methods are ill-suited to preserve the crucial structural semantics and contextual relationships within these network flows. This shortcoming directly impacts the Transformer's ability to effectively learn from and generalize across varied IoT traffic patterns.

In response to these identified challenges, this study introduces MIND-IoT, a novel and scalable hybrid framework combining transformer-based architectures with CNNs for IoT traffic classification. A key innovation is IoT-Tokenize, a custom tokenization pipeline that preserves the structural semantics of network flows. The study utilizes public IoT datasets (UNSW IoT Traces and MonIoTr) for pre-training and additional public datasets (IoT Sentinel, YourThings, and IoT-FCSIT) for evaluation. Rigorous evaluations across multiple classification tasks (binary IoT/non-IoT detection, IoT category recognition, and specific device identification) demonstrate the model's robustness, adaptability, and generalization, showing notable improvements over traditional methods even with limited labeled data.

The key contributions of this work are:

a. A hybrid model architecture integrating transformers and CNNs for enhanced IoT traffic classification;

b. The development of a pre-training model specifically for IoT network traffic, facilitated by the custom IoT-Tokenize pipeline, enables generalization across varied environments and

c. A comprehensive evaluation and fine-tuning process demonstrating the model's adaptability, robustness, and superior performance with limited labeled data.

The rest of this article is structured as follows: the Background section reviews relevant studies, the Methodology outlines the research methods, the Results and Discussion section presents our findings with detailed analysis, and the Conclusion summarises key insights and future research directions.

## BACKGROUND STUDIES

Traffic classification, a crucial process in networking, involves categorizing data packets based on various attributes to optimize network performance and management. This process includes analyzing traffic flow data, network topology data, configuration data, event logs, performance metrics, sensor data, and resource usage data (*Wijesekara & Gunawardena, 2023*). In the IoT, the efficient and accurate classification of network traffic has become increasingly important due to the rapid proliferation of IoT devices and the associated security risks. Early approaches, such as port-based and packet-based analysis, were limited in their ability to handle encrypted or obfuscated traffic (*Feizollah, Anuar & Salleh, 2018*). Subsequently, traditional machine learning techniques, such as decision trees, support vector machines, and random forests, offered incremental improvements through statistical feature analysis (*Zaki, Gani & Anuar, 2020*). For instance, *Chowdhury, Idris & Abas (2024)* demonstrated a lightweight device fingerprinting (DFP) model using

five-packet statistical features and random forest, achieving high accuracy. However, such models relied on static, hand-engineered features, making them less adaptable to dynamic IoT environments. On the other hand, deep learning models like CNNs and RNNs have enabled automatic feature extraction but often struggle with capturing long-range dependencies and require large labelled datasets (*Hameed, Violos & Leivadeas, 2022*). Even though *Chowdhury, Idris & Abas (2023)* showed that high classification accuracy could be achieved using only two features and a small number of packets in a CNN framework, these models remained prone to overfitting, limiting their generalizability.

To address these limitations and the challenges of data scarcity, pre-training approaches emerged as a promising direction. These methods leverage large-scale unlabeled datasets to learn robust representations, thereby enhancing adaptability and reducing dependency on extensive labeled data. For instance, *Liao, Li & Zhang (2019)* demonstrated rapid adaptation of a CNN model through pre-training on raw packet data. Similarly, *Li et al. (2023)* developed a pre-trained model to recognize unknown encrypted traffic patterns. Nevertheless, these approaches continued to face challenges, including the substantial computational resources required during the fine-tuning phase and difficulties in generalizing across highly variable IoT contexts.

A significant shift in this landscape occurred with the introduction of transformer-based models. Initially a breakthrough in natural language processing (NLP), transformers are exceptionally capable of capturing long-range dependencies and contextual relationships (*Vaswani et al., 2017*). Building on these strengths, transformers have been successfully adapted for applications in computer vision, speech recognition, and, increasingly, network traffic analysis (*Yang et al., 2020*; *Huang et al., 2021*; *Ahmed, Kientz & Kashef, 2023*). Crucially, their emergence aligns well with the principles of pre-training established in earlier deep learning efforts, as transformers excel at learning generalized representations from large-scale, unlabeled data, thereby reducing the reliance on extensive labeled datasets. This capability is particularly relevant in the IoT context, where acquiring comprehensive, labeled datasets is often challenging due to privacy concerns, rapid device turnover, and the diverse range of services and protocols that exist.

Recent works have started to apply transformers to IoT traffic classification. For example, DDosTC (*Wang & Li, 2021*) and RTIDS (*Wu et al., 2022*) demonstrated the capability of transformers to detect complex IoT attack patterns. These models specifically benefit from pre-training mechanisms that enable them to generalize across various network environments and attack types. In recent works, ITCT (*Bazaluk et al., 2024*) utilized TabTransformer for improved classification of MQTT traffic under data scarcity, while O2CPRCG-TransNet applied optimization algorithms to enhance intrusion detection accuracy (*Muthulingam et al., 2024*). Models such as TNN-IDS focused on detecting malicious traffic in MQTT traffic to address class imbalance (*Ullah et al., 2023*). IoT-Portrait (*Wang, Zhong & Li, 2023*) adopted incremental learning to manage continual learning challenges, and CSformer (*Jia et al., 2024*) introduced architectural efficiencies to reduce computational and memory overhead. Although these contributions are promising, their scope remains narrow, often limited to single-purpose tasks. Additionally, existing tokenization techniques fall short in capturing the structural complexity of IoT traffic,

highlighting the need for customized tokenization solutions that are vital for accurate IoT traffic interpretation. This highlights the need for more generalized and adaptable transformer-based solutions that can address the broader complexities of IoT traffic classification.

To address these specific challenges, we introduce MIND-IoT, an integrated transformer-based model enhanced by CNN architectures specifically designed to address the diverse and dynamic nature of IoT network environments. This study primarily focuses on traffic flow data as the basis for classification. This decision aligns with our objective of capturing the full communication context between devices through bidirectional flows, which is crucial for precise device-level and behavioral classification in dynamic IoT environments. Central to our approach is IoT-Tokenize, a custom tokenization pipeline that preserves semantically critical attributes, such as flow duration and packet size, structuring them into feature-value pairs that maintain contextual integrity. This structured input enables the Transformer's self-attention mechanism to learn dependencies across features, overcoming the limitations of traditional methods that cannot infer such relationships. The hybrid design leverages robust pre-training on large, diverse datasets (*e.g.*, UNSW IoT Traces and MonIoTr) to build generalized representations, which are then fine-tuned for specific tasks. This reduces reliance on labeled data and enhances adaptability across varied IoT environments. By combining Transformers' global context modeling strength with CNN's local feature extraction, MIND-IoT achieves superior accuracy and robustness across multiple datasets and classification tasks, as demonstrated in our experimental benchmarks.

## METHODOLOGY

This study leverages a Transformer-based framework as the base model for IoT network traffic classification, incorporating innovative tokenization and pre-training processes. Our approach is distinct within the IoT domain, as we develop a base model that can be fine-tuned for specific network traffic classification tasks with small data. To further enhance the model's performance, we integrate a 1-dimensional CNN during the fine-tuning stage, optimizing the model for various classification levels. Figure 1 illustrates the overall process, which we term "MIND-IoT," and structured around four key modules:

a. *Data preparation*: This module is responsible for collecting and preprocessing datasets, ensuring that the data is clean, structured, balanced, and adequately prepared for subsequent learning processes.

b. *Tokenization pipeline*: MIND-IoT employs a custom vocabulary-driven tokenization technique, referred to as "IoT-Tokenize," to parse and tokenize network features. This process enables efficient handling and processing of diverse IoT traffic data, ensuring that the model can effectively learn from the wide variety of traffic patterns found in IoT networks.

c. *Pre-training*: The pre-training module constructs a robust representation model for network traffic, designed to capture the complex patterns and traces inherent in IoT

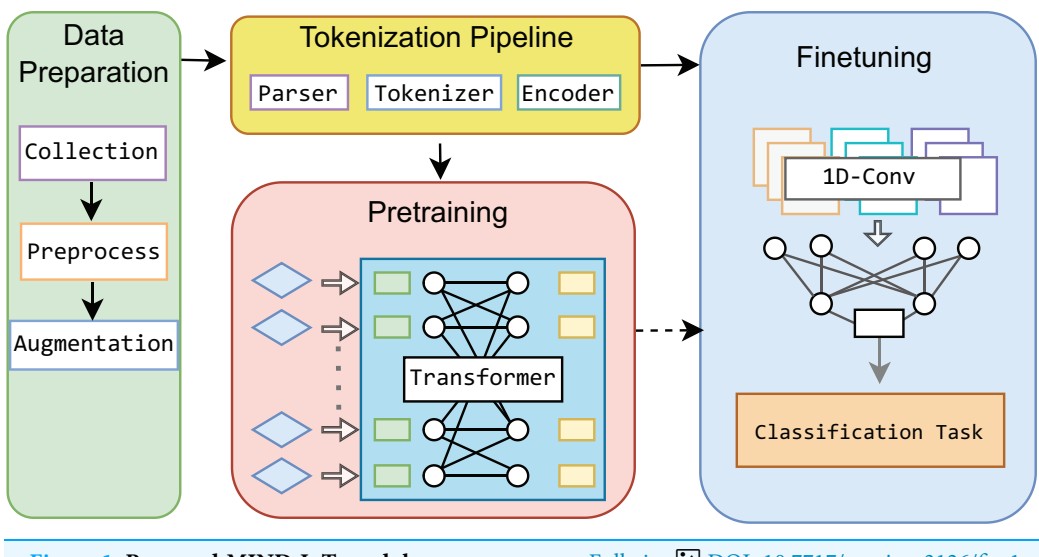

**Figure 1  Proposed MIND-IoT modules.**     

data. This foundational model, known as "IoT-Base," serves as the core of the MIND-IoT framework, enabling it to generalize well across various IoT tasks.

d. *Fine-tuning*: In the fine-tuning module, we refine the pre-trained IoT-Base model to address three specific classification tasks: distinguishing between IoT traffic and non-IoT traffic, categorizing IoT devices into relevant classes, and identifying specific device types. This approach enhances the model's versatility and accuracy across different classification levels, making it adaptable to various IoT network scenarios.

## Data preparation

The data preparation module collects, transforms, and preprocesses raw network traffic, optimizes the feature set, and addresses class imbalances to generate the dataset for tokenization. This process consists of three key steps: data collection, preprocessing, and augmentation.

### Data collection

In this study, we utilize two public datasets, UNSW IoT Traces and MonIoTr, for pre-training, as they provide large-scale, raw PCAP-format traffic collected from diverse smart home IoT devices. We selected these datasets strategically based on domain relevance, data volume, and format compatibility with our tokenization pipeline. They offer a rich and representative source for building generalized pre-trained models. To demonstrate the generalizability of our model with minimal fine-tuning, we include two additional public datasets, IoT Sentinel and YourThings, as well as one dataset collected in-house (IoT-FCSIT). These datasets vary in terms of network environments, device types, and data collection setups, thereby serving as robust benchmarks to validate the adaptability of our pre-trained model in different real-world scenarios.

a. *UNSW IoT Traces*. This widely used dataset is publicly accessible and includes traffic data from 23 unique IoT devices (*e.g.*, cameras, sensors, appliances) and several non-IoT devices (*Sivanathan et al., 2019*). Captured over 26 weeks, it offers insights into both autonomous and user-driven traffic, helping analyze IoT behavior in smart environments.

b. *MonIoTr*. This dataset focuses on privacy risks by analyzing information exposure from 46 IoT devices in 34,586 controlled experiments (*Ren et al., 2019*). Collected over a month, it covers traffic destination, encryption, and sensitive data exposure, contributing to privacy and security research in IoT.

c. *IoT Sentinel*. Introduced by *Miettinen et al. (2017)*, it covers the network traffic generated during the initial setup of 31 diverse smart home IoT devices. The dataset captures the traffic patterns for each device type, with at least 20 repetitions over four months. Despite being collected in 2017, this dataset remains a valuable resource due to the wide variety of IoT device types it includes, providing insights into the prevalent IoT traffic characteristics during that time.

d. *YourThings*. A dataset from the Georgia Institute of Technology was collected from 61 IoT and five non-IoT devices over 10 days (*Alrawi et al., 2019*). It aims to improve understanding of home IoT deployments and contribute to developing security measures.

e. *IoT-FCSIT*. We collected this dataset in the Faculty of Computer Science and Information Technology lab, Universiti Malaya, to evaluate the proposed classification model's robustness across various network environments (*Aqil et al., 2022*). It captures typical consumer behavior from six IoT and three non-IoT devices using a packet sniffer tool that mirrors the traffic passing through the Wi-Fi router. The data spans two months, from February 2022 to March 2022, and is available for access through the DOI: 10.6084/m9.figshare.25143581.v1.

### Data preprocessing

Data preprocessing converts raw network data into a format optimized for model training, ensuring consistency and relevance. This study uses network packets in PCAP format as input, and the process includes three key steps: (a) feature extraction, (b) data cleaning, and (c) data standardization. The result is a refined, standardized dataset that enhances the model's accuracy and efficiency for IoT traffic classification.

#### Feature extraction

We utilize NFStream, an open-source network analysis tool, to extract features from network traffic (*Aouini & Pekar, 2022*). NFStream aggregates packets into flows, calculating 85 flow features (*e.g.*, flow duration, total bytes, packets per flow) and outputting the results in a CSV file. This approach captures the full communication context between devices by focusing on bidirectional flows rather than individual packets, enabling precise device-level classification. This method is particularly useful for

differentiating devices with similar packet-level behaviors but distinct communication patterns.

*Data cleaning*

Data cleaning removes irrelevant features and unrelated flows to optimize the dataset. Non-informative features such as IP addresses and media access control (MAC) addresses are excluded. Likewise, flows related to network management (*e.g.*, address resolution protocol (ARP), dynamic host configuration protocol (DHCP), ICMP, NTP) are discarded to eliminate noise. After cleaning, 64 essential features remain, streamlining the dataset and boosting the model's effectiveness.

*Data standardization*

Standardization scales the data to enhance model performance. Continuous variables undergo Min-Max scaling from 0 to 1, ensuring all features contribute equally to model predictions. Instead of one-hot encoding, categorical variables are fed directly into the transformer model, leveraging the self-attention mechanism to dynamically assess their importance. This method improves the model's ability to capture relationships within categorical data.

These preprocessing steps, outlined in Table 1, ensure the dataset is comprehensive, unbiased, and normalized, providing a solid foundation for effective IoT traffic classification.

## Data augmentation

The augmentation process enhances the classification performance of underrepresented classes by generating synthetic data to balance the dataset, ensuring fair and effective model training across all classes. This data augmentation process is performed as a distinct step before the pre-training phase as part of the overall data preparation.

Our data augmentation approach is inspired by the work of *Chu & Lin (2023)*, which combines a multilayer perceptron (MLP) with generative adversarial networks (GANs). We specifically adopted their core concept of using a GAN-based augmentation for underperforming classes based on a predefined performance threshold. This methodology was adapted for IoT traffic data. Initially, the MLP acts as the primary classifier, processing preprocessed data. Training continues until the overall classification performance reaches a predefined threshold ($P_T$). To address individual class performance, a separate threshold ($P_i$) is set for each class. If the performance for a specific class $i$ falls below its threshold $P_i$, a class-specific GAN, $GAN\_C_i$ activates. $GAN\_C_i$ uses the original data from class $i$ to generate new synthetic instances ($K$), which are then integrated into the training set. This cycle repeats until both the overall performance threshold $P_T$ and the specific class performance threshold $P_i$ are met. If these criteria are satisfied, training concludes. Otherwise, the GANs continue to augment data for any class that has not yet met the set thresholds, thereby addressing gaps in the training data and enhancing the model's accuracy and robustness across all classes.

In this study, the data augmentation process successfully increased the number of data instances for underrepresented classes, resulting in a more balanced dataset. For example,
**Table 1 Data preprocessing steps.**

| Step | Description | Input | Output |
|------|-------------|-------|--------|
| Feature extraction | Extract statistical flow features from raw network packets. | Raw PCAP files | CSV file with 85 features per flow |
| Remove identifiers | Eliminate row and flow expiration identifiers to prevent bias. | 85 features, remove 2 | Data with 83 features |
| Remove IP and MAC addresses | Exclude IP, Port, and MAC addresses to maintain privacy and avoid overfitting to specific networks. | 83 features, remove 7 | Data with 76 features |
| Discard payload features | Remove payload inspection features to concentrate on flow characteristics. | Data with 76 Features, remove 12 | Data with 64 features |
| Handle missing values | Identify and remove rows with missing data to ensure dataset integrity. | Data with missing values | Clean data without missing values |
| Remove management packets | Remove management packets (*e.g.*, ARP, DHCP, ICMP) to focus on application traffic. | Data with management packets | Data without management packets |
| Normalise features | Apply Min-Max scaling to rescale numerical features uniformly. | Data with raw numerical values | Normalized data |

the UNSW IoT Traces dataset grew from 239,247 to 244,608 traffic flows, and the MonIoTr dataset expanded from 107,390 to 120,370 flows. Such augmentation is crucial for improving classifier performance, especially for minority classes, and prepares the data for subsequent tokenization and model training.

## Tokenization pipeline

Tokenization, a technique commonly used in NLP, is less prevalent in IoT network traffic classification. This study addresses this gap by introducing a novel approach that adapts tokenization techniques from NLP to enhance IoT network traffic classification. This study transforms traffic statistical features into a structured format suitable for neural networks by developing a custom tokenizer. This innovative approach applies tokenization to network traffic, enabling the model to learn internal representations while preserving essential syntactic and semantic information. The tokenization pipeline consists of three main components: (a) parser, (b) tokenizer, and (c) encoding. These components transform network traffic flow features into a format the neural network can effectively process.

### Parser

The parser transforms raw feature data into a structured sequence, maintaining the relationships between features and their values. It takes input from the data preprocessing module, which provides tabular data of 64 features. For each row, the parser generates a sequence of feature names paired with their values in the following format:

$$(\text{feature\_name}_1 \ \text{value}_1, \text{feature\_name}_2 \ \text{value}_2, \dots \ \text{feature\_name}_{64} \ \text{value}_{64},). \tag{1}$$

This format maintains both syntactic and contextual relations among features by keeping them sequentially and semantically coupled, enabling the model to learn dependencies (*e.g.*, correlations between flow duration and packet size) during pre-training. Figure 2 illustrates a sample of the transformed IoT traffic data in json.

**Figure 2 Sample of the transformed IoT traffic data.**

**Table 2 Custom pre-defined tokens.**

| Token type | Total | Examples |
|---|---|---|
| Reserved token | 5 | <pad>, <unk>, <mask>, <s>, </s> |
| Pre-defined token | 64 | 'ptcl', 'ipv', 'vln', 'tnnl', 'bi_dur', 'bi_pkt', 'bi_byte', 's2d_dur', 's2d', 's2d_byte', 'd2s_dur', 'd2s', 'd2s_byte', 'bi_min_ps', 'bi_mean_ps', 'bi_std_ps', 'bi_max_ps', 's2d_min_ps', 's2d_mean_ps', 's2d_std_ps', 's2d_max_ps', 'd2s_min_ps', 'd2s_mean_ps', 'd2s_std_ps', 'd2s_max_ps', 'bi_min_pi_ms', 'bi_mean_pi_ms', 'bi_std_pi_ms', 'bi_max_pi_ms', 's2d_min_pi_ms', 's2d_mean_pi_ms', 's2d_std_pi_ms', 's2d_max_pi_ms', 'd2s_min_pi_ms', 'd2s_mean_pi_ms', 'd2s_std_pi_ms', 'd2s_max_pi_ms', 'bi_syn', 'bi_cwr', 'bi_ece', 'bi_urg', 'bi_ack', 'bi_psh', 'bi_rst', 'bi_fin', 's2d_syn', 's2d_cwr', 's2d_ece', 's2d_urg', 's2d_ack', 's2d_psh', 's2d_rst', 's2d_fin', 'd2s_syn', 'd2s_cwr', 'd2s_ece', 'd2s_urg', 'd2s_ack', 'd2s_psh', 'd2s_rst', 'd2s_fin', 'req_server_name', 'client_fingerprint', 'server_fingerprint' |

### Tokeniser

The tokenizer converts these structured sequences into tokens that the model can process. This study employs byte pair encoding (BPE), adapted for IoT traffic, to reduce vocabulary size while capturing essential information. The tokenizer prepares inputs by breaking down feature sequences into finer-grained tokens.

(i) *Byte pair encoding.* BPE reduces the vocabulary size, optimizing it for the model's embedding layer. By tokenizing at the subword level, BPE minimizes the likelihood of out-of-vocabulary tokens. The tokenizer uses byte-level BPE from (*Radford et al., 2019*), fixing the BPE size at 256 to eliminate out-of-vocabulary tokens and setting the maximum vocabulary size to 52,000 entries. This balance helps control the number of trainable parameters, simplifying the learning process.

(ii) *Reserved tokens.* These tokens are unique tokens reserved for defining the input structure within the model. These include tokens like <s> (start of sequence), </s> (end of sequence), <pad> (padding), <unk> (unknown token), and <mask> (masking token). These tokens help maintain uniform sequence lengths and provide consistency when handling unknown or masked elements during training.

(iii) *Predefined tokens.* Predefined tokens, such as feature names, are excluded from subword tokenization to preserve their syntactic or semantic meaning. This study retains 64 predefined tokens that represent key network traffic features, as summarized in Table 2. Combining BPE with predefined tokens ensures efficient vocabulary building while retaining meaningful information about traffic flows.

*Encoding*

Encoding converts tokens into tensors, which are multidimensional arrays used in machine learning models. Each token is mapped to its corresponding index in the vocabulary, and a tensor is constructed based on sequence length and padding type. The maximum sequence length for pre-training is set at 512 to optimize learning across various devices. In contrast, the fine-tuning sequence length is extended to 1,024 to accommodate 90% of samples without truncation. Right-padding with reserved tokens ensures that all sequences maintain uniform length.

The output of the tokenization pipeline is a tensor representing the encoded sequence, which is ready for use in transformer-based models for both training and inference. This tokenization approach effectively handles IoT traffic data, ensuring that it is structured and encoded for optimal processing in the pre-training and fine-tuning modules.

## Pre-training

This study follows the transformer architecture similar to the original RoBERTa implementation (*Liu et al., 2019*), building on the foundational Transformer model introduced by *Vaswani et al. (2017)*. Although the original Transformer includes both encoder and decoder components, our approach utilizes only the encoder to develop more effective input embeddings for subsequent fine-tuning. The encoder processes the input sequence by converting it into a series of hidden states through multiple identical layers, with each layer containing two sub-layers: (1) a multi-head self-attention mechanism and (2) a fully connected feed-forward network.

### *Multi-head self-attention mechanism*

The self-attention layer allows the model to weigh the importance of different tokens in the input sequence relative to one another. For each token, it computes three vectors: Query ($Q$), Key ($K$), and Value ($V$). These vectors are obtained by multiplying the input embeddings with learned weight matrices:

$$Q = H^{(l-1)} W^Q, \quad K = H^{(l-1)} W^K, \quad V = H^{(l-1)} W^V \tag{2}$$

where $H^{(l-1)}$ represents the output from the previous layer $l$ (or the input embeddings for the first layer), and $W^Q, W^K, W^V$ are the learned projection matrices. In the first layer, where the embedding vector is denoted as $E'$ from encoding, $H^0 = E'$. Each layer $l$ performs self-attention and feed-forward operations. This mechanism allows each feature-value pair to attend to others in the sequence, thereby enabling the model to discover relationships, such as the correlation between packet length and flow duration, without manual encoding. The self-attention scores are computed as follows:

$$\text{Attention}(Q, K, V) = \text{softmax}\left(\frac{QK^T}{\sqrt{d_k}}\right) V. \tag{3}$$

Here, $Q$, $K$, $V$ are the linear projection of $H^{l-1}$ and $d_k$ is the dimension of the key vectors, which scales the dot products to prevent exceedingly large values in the softmax function.

Subsequently, the multi-head attention allows the model to focus on different parts of the input by applying the self-attention mechanism multiple times with different learned projections, concatenating the results, and projecting them again;

$$MultiHead(Q, K, V) = Concat(head_1, head_2, \dots, head_h)W^O \tag{4}$$

and each attention head is computed as follows:

$$head_i = Attention(QW_i^Q, KW_i^K, VW_i^V) \tag{5}$$

where $W_i^Q, W_i^K, W_i^V, W^O$ are the learned projection matrices. After the multi-head self-attention layer, the output passes through a feed-forward network, which is applied to each position separately and identically.

### Fully connected feed-forward network

The feed-forward network (*FFN*) consists of a two-layer neural network with a ReLU activation function in between.

$$FFN(x) = \max(0, xW_1 + b_1)W_2 + b_2 \tag{6}$$

where $W_1, W_2$ are weight matrices and $b_1, b_2$ are biases. Each of the sub-layers (self-attention and feed-forward) is followed by layer normalization (*LayerNorm*) and residual connections to ensure stable training. The output of the last encoder layer $H^l$ is then used for downstream tasks, with each encoder layer performing the following normalization:

$$Layer_l(H) = LayerNorm(H + FFN(SelfAttention(H))). \tag{7}$$

### Learning objective

In this study, the MIND-IoT pre-training modules employ a self-supervised learning approach as the primary learning objective. This learning objective helps uncover latent patterns and structures inherent in the IoT network traffic data without requiring extensive manual labeling. Specifically, the self-supervised task leverages the masked language modeling (MLM) objective introduced by *Devlin et al. (2018)*. During pre-training, the MLM objective guides the model in predicting masked tokens within input sequences, thereby enabling it to acquire informative representations for IoT traffic statistical flow features.

The MLM objective operation trains the model by masking some input tokens and calculates the probability of predicting the original token $x_i$ given the masked input $X_{masked}$. The loss function for MLM is calculated as follows:

$$Loss_{MLM} = -\sum\nolimits_{masked \ x_i} \log P(x_i | X_{masked}). \tag{8}$$

This objective helps the model learn contextualized representations by requiring it to predict missing tokens based on their context.

### Experiment setup and configuration

This study utilized PyTorch 1.9 with CUDA 11.0 for the pre-training experiments, running on Python 3.9. The training was conducted on a high-performance A2 series virtual

machine provided by Google Cloud. On the other hand, the fine-tuning phase was conducted on a local Windows 11 desktop. This machine features a 12-core Intel i7 Gen processor, 32 GB of RAM, and an Nvidia GeForce RTX 3090 GPU with 24 GB of video memory. While less powerful than the cloud-based setup used for pre-training, this configuration was still sufficient for fine-tuning tasks, allowing for efficient model adaptation and testing.

### Pre-train model specification

The pre-training involved training a 6-layer transformer encoder with masked self-attention heads, each having 768-dimensional states and 12 attention heads. The position-wise feed-forward networks use 3,072-dimensional inner states with GELU activation functions. The model's embedding layer includes word embeddings with a vocabulary size of 52,000, position embeddings with a maximum length of 576, and token-type embeddings, followed by layer normalization and 0.1 dropout for regularization.

We customized the tokenizer using the Byte-Level BPE tokenizer, trained on balanced data from previous modules with a vocabulary size of 52,000 and a minimum frequency threshold of 2. The training process used the Adam optimization scheme with gradient accumulation over eight steps, effectively creating a batch size of 128 sequences. We trained the model for 500 epochs, each epoch consisting of mini-batches of 16 sequences. Standard practices from previous RoBERTa implementations were followed for learning rate scheduling and regularisation, ensuring effective convergence and generalization.

For data preprocessing, the custom tokenizer was configured with a maximum length of 576 tokens, handling both padding and truncation. We trained the language model with a masked language modeling probability of 0.15. The final classification head consisted of a dense layer with 768-dimensional inputs, a dropout layer with a dropout rate of 0.1, and an output projection layer, depending on the type of classification task. The training process included regular checkpoints every 500 steps, with a limit of keeping only the last five checkpoints to manage storage. The entire training module was designed to leverage GPU acceleration, ensuring efficient and scalable model training.

## Fine-tuning

After pre-training, we fine-tuned the model to evaluate its performance across specific classification tasks. This process utilized the pre-trained embeddings as initial weights and further trained them on labeled datasets, adapting the model to the target tasks, (Fig. 3). We implemented two classification approaches during fine-tuning: a standard MLP and a CNN. The MLP used the complete pre-trained model, while the CNN, known for its efficiency and speed, leveraged its robust architecture for faster fine-tuning.

### MIND-MLP

We applied a fully connected layer with 768 neurons and an output layer with 2 or 41 neurons, depending on whether the task was binary or multi-class classification. The pre-trained MIND-IoT weights were reused, and fine-tuning continued for several epochs.

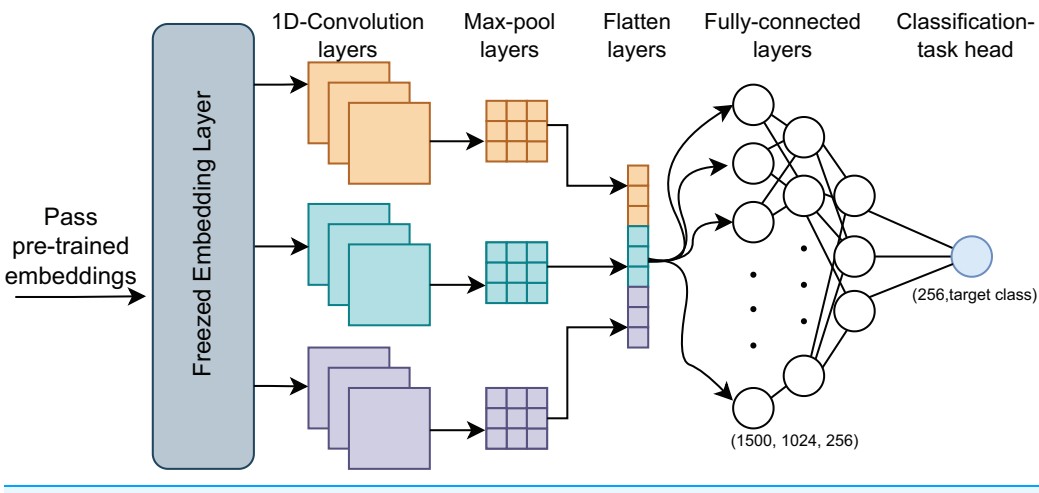

**Figure 3 MIND-IoT fine-tuning structure.**

This approach, widely used in pre-trained model fine-tuning, requires minimal changes to the MIND-IoT architecture.

### MIND-CNN

We used the RobertaClassificationHead to handle three distinct tasks: binary classification of IoT *vs.* non-IoT traffic, IoT category classification, and IoT device classification.

(i) Task 1: Binary classification (IoT *vs.* Non-IoT Traffic)
The model differentiated IoT from non-IoT traffic using a dense layer (768 features), a dropout layer (0.1 probability), and a final output layer for binary classification. This task is foundational for identifying IoT traffic in broader network environments.

(ii) Task 2: IoT category classification
In this task, the model classified traffic into one of six IoT categories: surveillance, entertainment systems, smart hubs, appliances, environmental control, and energy management. The architecture mirrored the binary task, featuring an output layer for six classes, which provided detailed traffic classification within IoT domains.

(iii) Task 3: IoT device classification
The model identified the specific IoT device generating the traffic, with class sizes varying by dataset (*e.g.*, 23 classes for UNSW IoT Traces, 46 for MonIoTr). This task enabled precise device-level classification, which is crucial for managing security and device-specific optimizations.

The fine-tuning dataset included diverse IoT traffic samples representing various scenarios. The dataset was split into 80% training and 20% test sets. We optimized hyperparameters such as learning rate, batch size, and epochs, starting with a learning rate of 2e−5 and dynamically adjusting it using a learning rate scheduler. Early stopping was implemented to avoid overfitting. Lastly, the key evaluation metrics included accuracy, F1-score, precision, and recall, with a particular focus on the F1-score to strike a balance between precision and recall.

## Assessment metrics

This study evaluates the MIND-IoT model using four standard performance metrics: accuracy, F1-score, precision, and recall. These metrics are essential for assessing the model's overall performance in IoT traffic classification tasks:

(i)  Accuracy = TP + TN/(TP + TN + FP + FN)

(ii)  F1 = 2 × (Precision × Recall)/(Precision + Recall)

(iii)  Precision = TP/(TP + FP)

(iv)  Recall = TP/(TP + FN)

where TP, TN, FP, and FN represent true positives, true negatives, false positives, and false negatives, respectively.

## RESULTS AND DISCUSSION

This section evaluates the IoT-Base model in three parts. First, the pre-trained model is assessed using three configurations (large, medium, and small) to gauge performance and analyze the effect of dataset size on learning outcomes. Next, the model's classification ability is independently tested for each task, including IoT *vs*. non-IoT, IoT category, and IoT device type, providing detailed insights into its effectiveness and validating the proposed tokenizer. Lastly, a benchmark evaluation using three publicly available datasets highlights the model's robustness and compares its performance to that of other state-of-the-art methods in IoT traffic classification.

### Pretrained model evaluation

This study evaluates the pre-trained MIND-IoT model, which leverages MLM as its primary pre-training objective. For pre-training, we exclusively utilized the UNSW IoT Traces and MonIoTr datasets, as the presence or nature of labels does not impact this phase of learning.

To understand the influence of model parameters on pre-training performance, we systematically investigated three distinct pre-training configurations: small, medium, and large. Table 3 details the specific hyperparameters used for each of these configurations. Each pre-training session typically spanned 240 to 366 h (equivalent to 10 to 14 days), with variations depending on the model configuration. Regarding the training duration, this study pre-trains three models with different hyperparameter configurations. Pre-training was conducted for over 15,000 steps, with training stopped at approximately 20,000 steps across all configurations to prevent overfitting as the training loss began to plateau.

Table 4 presents the recorded pre-training loss for each model configuration. The small configuration yielded a training loss of 0.9366 after 20,000 steps. When the model size was increased to the medium configuration, the training loss significantly decreased to 0.6944, representing a 34.88% improvement. Subsequently, the large configuration showed a further, albeit marginal, improvement of 3.37%. Beyond this point, further increases in model size did not result in additional performance improvements.

**Table 3 Hyperparameters of pre-train configurations.**

| Configuration | No. of layers | Encoder dimension | Feed-forward network size | Attention heads | Total parameters |
|---|---|---|---|---|---|
| Large | 12 | 768 | 3,072 | 12 | 124,836,866 |
| Medium | 6 | 768 | 3,072 | 12 | 82,359,632 |
| Small | 3 | 768 | 3,072 | 3 | 61,096,016 |

**Table 4 Pretraining results for different model configurations.**

| Config | Total batch size | Training loss (20 K Steps) |
|---|---|---|
| Large | 24 | 0.6710 |
| Medium | 32 | 0.6944 |
| Small | 44 | 0.9366 |

Figure 4 visually validates these findings, illustrating that as the number of model parameters increases, the training loss generally decreases. However, a key observation is the barely discernible difference in training loss between the medium and large configurations, contrasting sharply with the substantial improvement seen from the small configuration. Furthermore, Fig. 4 reveals a steep initial decrease in training loss across all configurations during the first 5,000 training steps. Conversely, after approximately 15,000 steps, the training loss for all configurations began to plateau, signalling the onset of overfitting to the training set. Consequently, pre-training was halted at this juncture (around 20,000 steps) to mitigate further overfitting.

Based on this comprehensive analysis, the medium configuration was selected as the reference pre-trained model for subsequent tasks. While the large configuration demonstrated a slightly better training loss, the medium configuration offered a significant advantage by having fewer total parameters. This reduction in parameter count contributes to improved training time, making the medium configuration a more efficient choice while consistently delivering robust performance.

### Pre-training data evaluation

Beyond model size, evaluating the impact of increasing training data size on specific task performance is crucial for pre-training models. The MIND-IoT model was pre-trained using various dataset combinations. Subsequently, these models were fine-tuned for a categorical classification task using *RobertaClassificationHead*, as not all datasets may have the same unique devices. However, they can consistently be defined within the same IoT categories. Leveraging the consistent definition of IoT categories across diverse datasets, irrespective of unique device types.

Table 5 presents the validation F1-scores for different dataset combinations used in both pre-training and fine-tuning. Results show that models pre-trained with UNSW IoT Traces, MonIoTr, and IoT Sentinel significantly improve performance when validated on these datasets. Specifically, the UNSW IoT Traces dataset experiences substantial enhancement when combined with the MonIoTr and IoT-FCSIT datasets. In contrast, the inclusion of IoT Sentinel in pre-training yields notable improvements for both the IoT

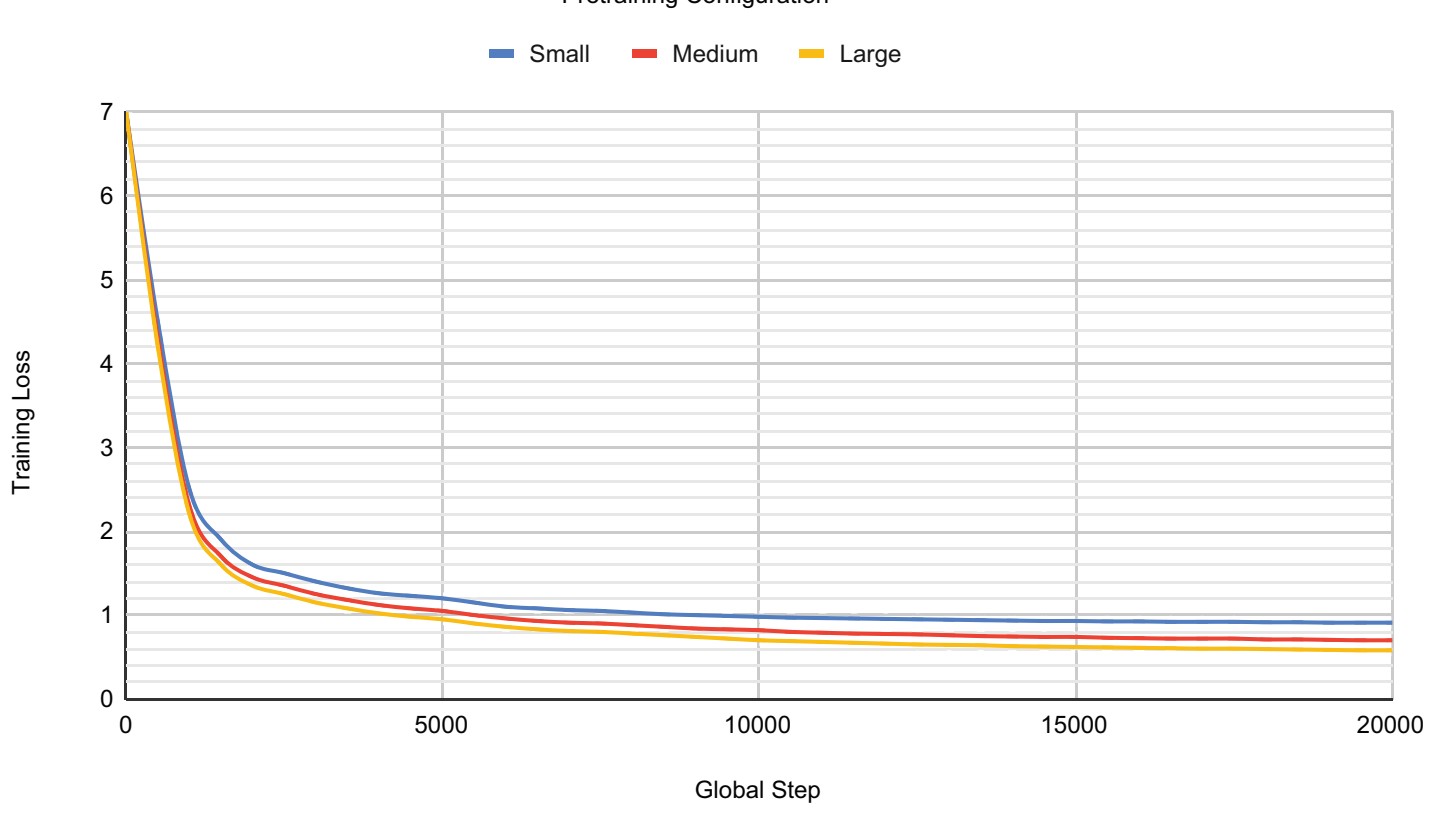

**Figure 4 Pretraining loss for different model configurations.**

**Table 5 Result on MIND-IOT MLP with different dataset pre-trained.**

| Pretrain & Finetune | Validation F1-score (%) | | | |
|---|---|---|---|---|
| | **UNSW** | **MonIoTr** | **Sentinel** | **IoT-FCSIT** |
| UNSW | 94.48 | 65.77 | 53.84 | 66.47 |
| UNSW, MonIoTr | 94.53 | 95.34 | 79.21 | 88.19 |
| UNSW, MonIoTr, Sentinel | 94.51 | 95.42 | 97.13 | 88.26 |

Sentinel and IoT-FCSIT validation sets. Conversely, UNSW IoT Traces do not exhibit substantial enhancement when IoT Sentinel is included in the pre-training, except when paired with MonIoTr and IoT-FCSIT, suggesting that IoT Sentinel lacks distinct features that are representative or beneficial for tasks specific to UNSW IoT Traces.

The result indicates that models pre-trained with additional datasets generally perform better. Specifically, models pre-trained with UNSW IoT Traces, MonIoTr, and IoT Sentinel show improved performance across all validation sets, particularly for IoT Sentinel and IoT-FCSIT. This result suggests that including more varied data in pre-training can lead to a model that better generalizes new, unseen data. However, the marginal improvement from adding IoT Sentinel to a model already trained on UNSW

IoT Traces and MonIoTr suggests that IoT Sentinel might contribute less significantly unique or representative features compared to IoT-FCSIT.

Given these observations, this study pre-trains with UNSW IoT Traces and MonIoTr for subsequent tasks. The decision relies on the substantial volume of data in UNSW IoT Traces and MonIoTr, which contain more than 300,000 data points. The results demonstrate the beneficial impact these datasets have on the model's ability to classify unseen data effectively. This approach strategically focuses on leveraging the most representative and beneficial data to enhance the model's performance, thereby optimizing computational resources and potentially mitigating the risks of overfitting associated with less representative data. This evaluation also demonstrates the pre-training model's ability to transfer knowledge between categories of IoT data, even when it is not trained in the same data environment. By using diverse datasets, the model learns more generalized features, thereby enhancing its ability to handle a variety of IoT traffic patterns and improving overall robustness in real-world applications.

## Classification evaluation

The classification evaluation focuses on fine-tuning and training individual tasks separately to provide granular insights into the model's performance. This approach isolates and evaluates the effectiveness of the pre-trained model for each classification task (binary IoT *vs.* non-IoT, category classification, and device type classification). By incorporating a single task head into the evaluation, the study tests and verifies the effectiveness of the proposed tokenizer and pre-training model across multiple tasks. The comparison models included three fine-tuning MIND-IoT variants: *MIND-Roberta* (using *a Roberta classification head*), *MIND-CNN* (with a CNN classification head), and *MIND-MLP* (featuring an MLP classification head). Additionally, three widely used learning models from previous IoT traffic classification research were employed: random forest (RF) (*Shahid et al., 2018*), long short-term memory (LSTM) (*Thapa & Duraipandian, 2021*), and convolutional neural network (CNN) (*Liao, Li & Zhang, 2019*). All classification models were trained and fine-tuned on the same feature set using the UNSW IoT Traces and MonIoTr datasets but evaluated across different tasks. Crucially, during the fine-tuning phase, the pre-trained MIND-IoT models were adapted and evaluated on additional unseen datasets (IoT Sentinel, YourThings, and IoT-FCSIT) to demonstrate their generalization capabilities. Each experimental result highlights the highest score for each performance metric in bold.

### Binary IoT and non-IoT classification task

In the first classification task, this study tests each model with three datasets: UNSW IoT Traces, MonIoTr, and IoT-FCSIT. UNSW IoT Traces and IoT-FCSIT include both IoT and non-IoT categories, while MonIoTr includes only IoT devices. This setup evaluates variations in network conditions across different datasets.

Figure 5 presents the F1-scores for the binary classification tasks of IoT and non-IoT traffic. The results demonstrate that the pre-trained MIND-CNN model achieves the

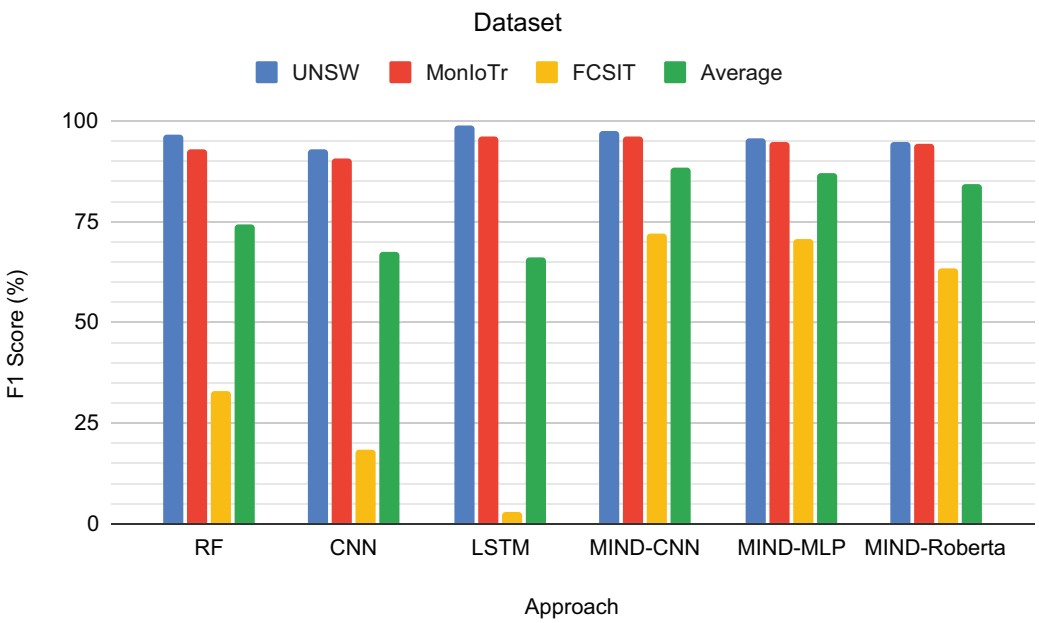

**Figure 5  Result for IoT and Non-IoT traffic classification.**

highest average F1-score of 88.46%, indicating robust performance across all datasets. MIND-MLP follows closely with an average F1-score of 86.90%, and MIND-Roberta achieves an average score of 84.10%. These results highlight the effectiveness of the MIND-IoT models, particularly MIND-CNN, in distinguishing IoT from non-IoT traffic.

The Random Forest model also performs well on the UNSW and MonIoTr datasets but shows a significant drop in performance on the IoT-FCSIT dataset, resulting in an average F1-score of 74.18%. Similarly, the CNN model exhibits moderate performance with an average F1-score of 67.20%. In contrast, the LSTM model shows high performance on UNSW and MonIoTr but fails on the IoT-FCSIT dataset, resulting in an overall average of 65.89%.

MIND-CNN captures complex patterns in the data, enabling it to distinguish between IoT and non-IoT traffic and achieve superior performance. The high performance of MIND-MLP and MIND-Roberta also validates the robustness of the MIND pre-training approach due to the transformer framework's ability to transfer learning to unseen or untrained data with the same structure and concept. The lower performance of traditional models, such as RF, CNN, and LSTM, on the IoT-FCSIT dataset suggests that these models may struggle with the variability and complexity of different IoT environments, underscoring the advantage of pre-trained models in handling diverse IoT traffic conditions.

Overall, these findings support the effectiveness of the MIND-IoT pre-trained models for binary IoT and non-IoT classification, demonstrating their potential to generalize well across various datasets and network conditions. This evaluation sets a solid foundation for exploring multi-class classification in subsequent experiments.

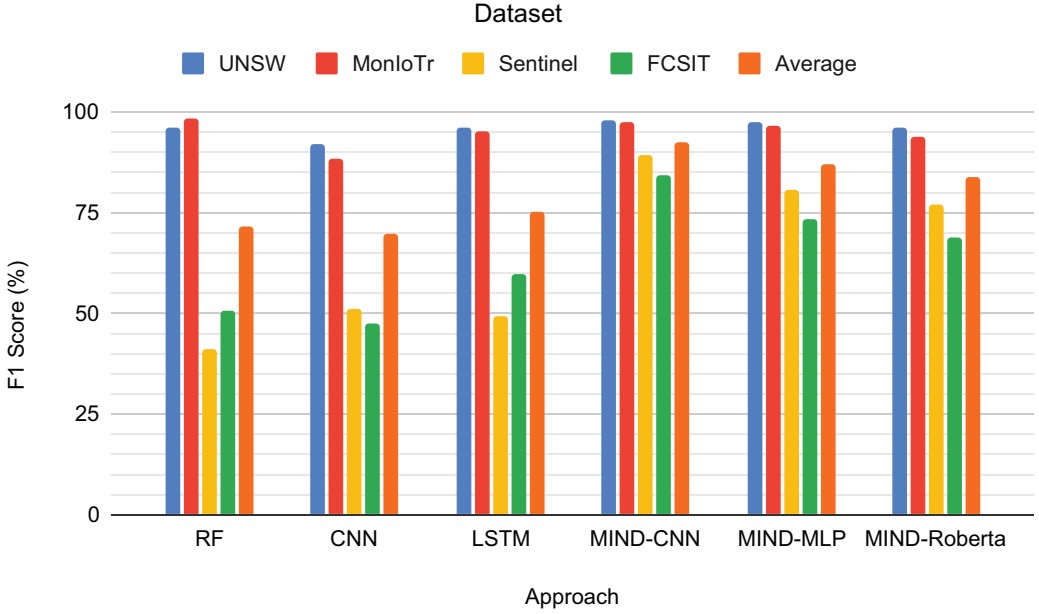

**Figure 6 Result for the category of IoT traffic classification.**

### IoT category classification task

The IoT category classification task evaluates the model's ability to classify IoT traffic into predefined categories. The evaluation used the same set of models, measuring their performance across different datasets.

Figure 6 shows the F1-scores for the IoT category classification task. The MIND-CNN model achieves the highest average F1-score of 92.21%, demonstrating its strong capability to classify IoT traffic into various categories accurately. This high performance reflects the model's deep learning architecture, which captures intricate patterns within the IoT traffic data. The MIND-MLP model also performs robustly, with an average F1-score of 86.89%, followed by MIND-Roberta, with an average score of 83.90%.

Traditional models, such as RF and LSTM, also demonstrate good performance, particularly on the UNSW and MonIoTr datasets, with average F1-scores of 71.52% and 75.09%, respectively. However, their performance drops significantly on the IoT Sentinel dataset, indicating that these models may struggle with the variability and complexity of different IoT categories. The CNN model, while effective, shows the lowest average performance among the tested models, with an average F1-score of 69.71%.

The superior performance of the MIND-CNN model highlights the effectiveness of the MIND-IoT pre-training approach, which leverages the transformer framework to transfer learning to unseen and untrained data with the same structure or concept. This capability enables the model to generalize more effectively across various IoT categories, ensuring consistent and accurate classification performance.

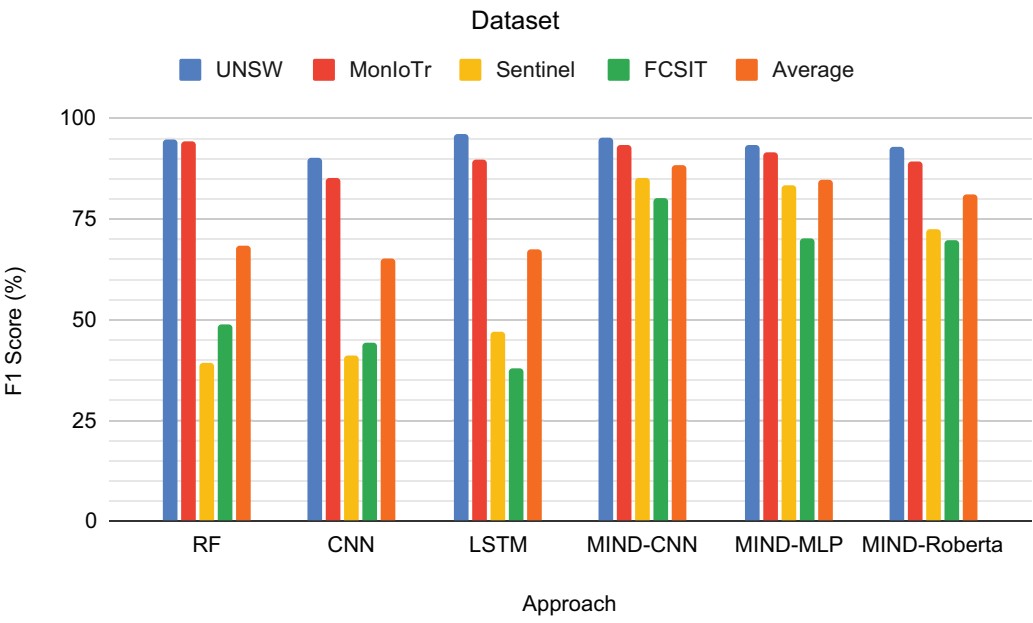

**Figure 7  Result for the device type of IoT traffic classification.**

### Device type classification task

The third task involves classifying IoT traffic by device type, with the number of classes varying depending on the dataset (UNSW: 23, MonIoTr: 46, Sentinel: 5, IoT-FCSIT: 3). The sentinel and IoT-FCSIT classes for the device classes are only 5 and 3 because this is the only device with the same type in the training dataset (UNSW and MonIoTr).

Figure 7 presents the F1-scores for the IoT device type classification task across different approaches and datasets. RF achieves high scores for UNSW (94.71%) and MonIoTr (94.22%) but performs poorly on Sentinel (39.13%) and IoT-FCSIT (48.58%). This result implies that while RF handles the larger datasets well, it struggles with the smaller and less diverse datasets. CNN shows similar patterns, with a solid performance on UNSW (90.36%) and MonIoTr (85.12%) but lower scores on Sentinel (41.26%) and IoT-FCSIT (44.41%). This result suggests that CNNs generally struggle to generalize from larger training sets to smaller, distinct validation sets.

Similarly, LSTM performs well on UNSW (95.98%) and MonIoTr (89.57%) but significantly worse on Sentinel (46.84%) and IoT-FCSIT (37.84%). Despite its strong sequence modelling capabilities, LSTM struggles with the variability and smaller size of Sentinel and IoT-FCSIT datasets. Conversely, MIND-CNN exhibits high performance across all datasets, particularly Sentinel (85.20%) and IoT-FCSIT (79.93%), reflecting its robustness and ability to generalize from varied data sources. Its strong performance on UNSW (95.26%) and MonIoTr (93.14%) further solidifies its effectiveness.

MIND-CNN emerges as the top performer with an average F1-score of 88.38%, indicating its superior ability to handle diverse datasets and generalize well across different

device types. MIND-MLP achieves an average score of 84.49%, demonstrating its effectiveness and adaptability in classifying various types of IoT devices. MIND-Roberta achieves an average score of 81.04%, demonstrating robust performance and effective knowledge transfer capabilities of the transformer framework. Traditional models, such as RF (68.16%), LSTM (67.56%), and CNN (65.29%), exhibit lower average scores, indicating their limitations in handling variability and generalizing from training to validation datasets.

The observed drop in performance for models like RF, CNN, and LSTM on unseen datasets can be attributed to several factors as follows:

a. *Random Forest*: RF constructs multiple decision trees during training and outputs the mode of the classes for the classification of each individual tree. The strength of RF lies in its ability to handle varied features and reduce overfitting by averaging results. This is consistent with the general understanding that tree-based machine learning methods often outperform neural network-based methods on tabular data (*Huang et al., 2020*). However, its performance drop on unseen datasets, such as Sentinel and IoT-FCSIT, suggests that the rules and splits created by RF highly depend on the specific characteristics and distributions of the training data. RF struggles to adapt when faced with new data that do not conform to these learned patterns, indicating a limited capacity for generalization beyond the training dataset.

b. *CNN and LSTM*: Neural network models, including CNNs and LSTMs, are powerful tools for capturing intricate patterns in data through deep learning. However, they are also prone to overfitting, especially when trained on a limited dataset. Overfitting occurs when the model learns the noise and details in the training data to the extent that it negatively impacts the performance of new, unseen data. This overfitting is evident in the substantial performance drop on the Sentinel and IoT-FCSIT datasets. These models may have become too specialized to the training data (UNSW and MonIoTr) and consequently failed to generalize well to datasets with different characteristics and distributions.

c. *Proposed pretrained models*: The proposed pre-trained models, particularly those utilizing the transformer framework (*e.g.*, MIND-Roberta), address these shortcomings by enabling the transfer and sharing of knowledge across datasets. The pre-training phase exposes the models to a broader array of data, allowing them to learn more generalized features and patterns. This pre-training approach builds a more robust foundation that can effectively handle variability and generalize to unseen data. By leveraging the shared structure and concepts across different datasets, the pre-trained models mitigate the risk of overfitting and enhance their adaptability to new, diverse datasets.

This capability to transfer learning and share knowledge between datasets is crucial in the context of IoT traffic classification, where the data can be highly varied and dynamic. The superior performance of the MIND-IoT models, particularly MIND-CNN and MIND-Roberta, underscores the value of this approach, offering a more resilient

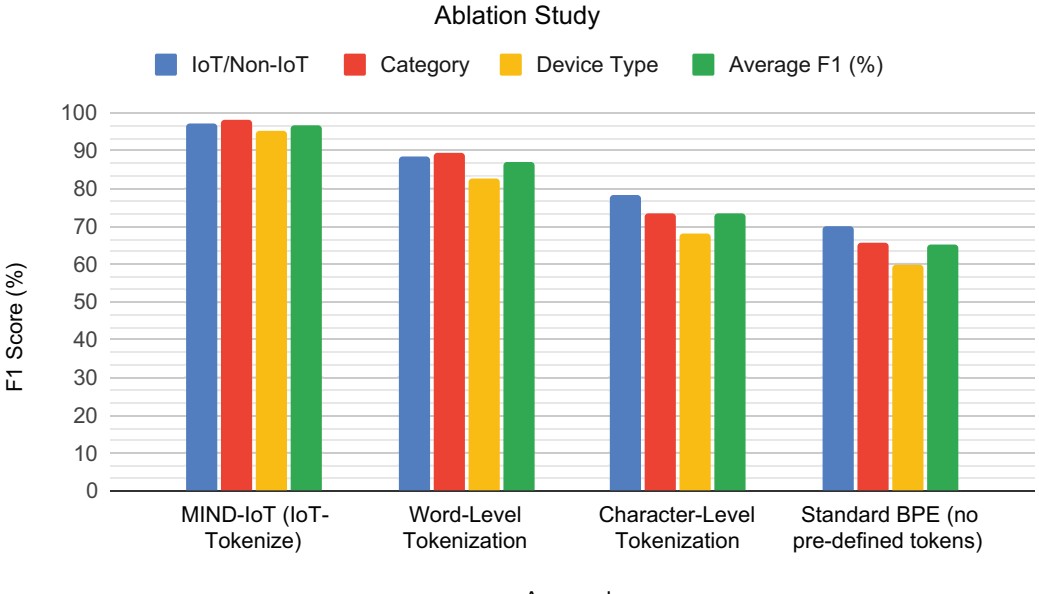

**Figure 8 Tokenization ablation study.** F1-scores for the different tokenization approaches across the binary IoT/Non-IoT, IoT category, and IoT device type classification tasks, along with their average performance.               

and effective solution for classifying IoT traffic across diverse environments and device types.

## Ablation study

To evaluate the critical role of our custom IoT-Tokenize pipeline, an ablation study was conducted to compare its performance with that of several standard tokenization strategies. This study aimed to isolate the impact of our domain-specific tokenization choices on the model's classification accuracy across different tasks. All models in this ablation were fine-tuned using the medium configuration and evaluated on the UNSW IoT Traces dataset, adhering to the same training and evaluation protocols previously described.

The comparison included:

a. Word-level tokenization: Each "feature_name value" pair was treated as a single, indivisible token.

b. Character-level tokenization: The input sequence was broken down into individual characters.

c. Standard BPE (without predefined tokens): Byte Pair Encoding was applied to the raw feature-value sequence without explicitly preserving feature names as predefined tokens.

To visually illustrate the impact of different tokenization strategies on model performance, a grouped bar chart is presented in Fig. 8. The results demonstrate the superior performance of our IoT-Tokenize approach across all classification tasks.

Word-level tokenization, while conceptually simple, showed a noticeable drop in F1-scores, indicating that merely treating feature-value pairs as atomic units is insufficient for capturing the intricate patterns within IoT traffic. Character-level tokenization yielded moderate performance, highlighting the critical loss of semantic meaning that occurs when feature names are fragmented into individual characters. Furthermore, applying standard BPE without incorporating our carefully selected predefined tokens performed the worst. This result highlights its inability to effectively learn from the structured nature of IoT traffic when specific domain knowledge, such as predefined feature names, is not explicitly preserved.

This ablation study validates the necessity and effectiveness of IoT-Tokenize. By structuring IoT traffic data into semantically meaningful feature-value pairs and leveraging predefined tokens, our custom tokenizer enables the Transformer model to capture complex syntactic and contextual relationships more effectively. This specialized tokenization is crucial for enhancing the model's ability to learn robust representations and generalize effectively across various IoT classification tasks. The following section will focus on benchmarking the proposed model against other related works, further validating its effectiveness in diverse IoT traffic classification scenarios.

## Benchmarking

In this section, the performance of the MIND-IoT pre-trained model is benchmarked against existing works to validate its effectiveness in various IoT classification tasks. The benchmarking process includes two key evaluations: (a) evaluation with IoT device type classification and (b) evaluation with manufacturer group classification.

## Evaluation with IoT device type classification

To further evaluate MIND-IoT pre-trained's performance, this study compares IoT device type classification task against three prior works that utilized the UNSW IoT Traces dataset: (a) *Sivanathan, Gharakheili & Sivaraman (2020)* using unsupervised one-class clustering, (b) *Chowdhury et al. (2020)* adopting device fingerprinting and training a PART classifier (a decision list-based algorithm), and (c) *Ali et al. (2021)* using statistical features derived from feature importance methods and training six machine learning models: random forest, decision tree, support vector machine (SVM), naïve Bayes, KNN, and AdaBoost. Additionally, *Ali et al. (2021)* also evaluated their models with a different dataset (trained with UNSW and evaluated with YourThings data).

This benchmark replicates all previous works to ensure a fair comparison by training with UNSW data and evaluating nine devices from the YourThings dataset, all of which have the same types as those in the UNSW classes. It is essential to note that the comparison was based on the reported results in publications marked as original (*ori*) and those with repeated experiments (*rep*). The evaluation focused on accuracy and F1-score, as these metrics are consistently reported across all compared works.

Table 6 compares the MIND-IoT models (MIND-CNN and MIND-MLP) with existing models from previous studies on device type classification. The results show that MIND-CNN and MIND-MLP consistently outperform the models from previous works,

**Table 6 Comparisons of MIND-IoT with existing works on device type classification.**

| Approach | Branch | | Dataset | | | |
|---|---|---|---|---|---|---|
| | | | UNSW | | YourThings | |
| | | | Acc. % | F1 % | Acc. % | F1 % |
| This work | MIND-CNN | – | 95.26 | 94.18 | 92.92 | 91.40 |
| | MIND-MLP | – | 93.41 | 91.33 | 86.11 | 87.09 |
| *Ali et al. (2021)* | RF | (Ori) | 99.00 | 80.00 | 79.00 | 81.00 |
| | | (Rep) | 96.01 | 91.05 | 63.82 | 62.47 |
| | DicisionTree | (Ori) | 69.00 | 61.00 | 72.00 | 63.00 |
| | | (Rep) | 71.38 | 65.94 | 66.71 | 62.82 |
| | SVM | (Ori) | 89.00 | 81.00 | 63.00 | 69.00 |
| | | (Rep) | 92.57 | 90.43 | 51.91 | 53.49 |
| | NaiveBayes | (Ori) | 85.00 | 89.00 | 85.00 | 89.00 |
| | | (Rep) | 83.05 | 84.82 | 65.11 | 62.80 |
| | KNN | (Ori) | 77.00 | 78.00 | 63.00 | 69.00 |
| | | (Rep) | 81.16 | 79.63 | 60.42 | 62.36 |
| | AdaBoost | (Ori) | 91.00 | 91.00 | 88.00 | 83.00 |
| | | (Rep) | 93.49 | 93.33 | 72.91 | 71.20 |
| *Sivanathan et al. (2019)* | Clustering | (Ori) | 97.00 | – | – | – |
| | | (Rep) | 96.51 | 92.01 | 62.77 | 60.32 |
| *Chowdhury et al. (2020)* | PART | (Ori) | 99.93 | – | – | – |
| | | (Rep) | 97.82 | 93.99 | 35.84 | 42.55 |

especially when evaluated on the YourThings dataset. MIND-CNN achieves an accuracy of 95.26% on UNSW and 92.92% on YourThings, with corresponding F1-scores of 94.18% and 91.40%. MIND-MLP also demonstrates strong performance, achieving accuracies of 93.41% on UNSW and 86.11% on YourThings, as well as F1-scores of 91.33% and 87.09%. These results highlight the robustness and generalization capability of the MIND-IoT models across different datasets.

In contrast, the replicated results of *Ali et al.*'s *(2021)* models show a notable drop in performance on the YourThings dataset compared to the original UNSW results. For example, the replicated RF model achieves 96.01% accuracy and 91.05% F1-score on UNSW but only 63.82% accuracy and 62.47% F1-score on YourThings. Similar performance drops are observed for decision tree, SVM, naïve Bayes, KNN, and AdaBoost. This drop in performance suggests that traditional machine learning models struggle to generalize to unseen data, likely due to overfitting the specific characteristics of the training dataset.

The unsupervised one-class clustering method by *Sivanathan et al. (2019)* and the PART classifier by *Chowdhury et al. (2020)* also exhibit significant performance drops in the repeated experiments on YourThings. The clustering approach's F1-score decreases from 92.01% on UNSW to 60.32% on YourThings, and the PART classifier's F1-score drops from 93.99% on UNSW to 42.55% on YourThings. These results further emphasize the limitations of these methods in handling diverse and unseen datasets.

**Table 7 Manufacturer groups in IoT sentinel.**

| No. | Manufacturer | Devices |
|---|---|---|
| 1 | Fitbit | Fitbit Aria WiFi-enabled scale |
| 2 | D-Link | D-LinkCam, D-LinkDayCam, D-LinkDoorSensor, D-LinkHomeHub, D-LinkSensor, D-LinkSiren, D-LinkSwitch, D-LinkWaterSensor |
| 3 | Edimax | EdimaxCam1, EdimaxPlug1101W, EdimaxPlug2101W |
| 4 | Ednet | EdnetCam1, EdnetGateway |
| 5 | HomeMatic | HomeMaticPlug |
| 6 | Hue | HueSwitch, HueBridge |
| 7 | Smarter | iKettle2, SmarterCoffee |
| 8 | Lightify | Osram Lightify Gateway |
| 9 | MAXGateway | Home automation sensors |
| 10 | TP-Link | TP-LinkPlugHS100, TP-LinkPlugHS110 |
| 11 | WeMo | WeMoInsightSwitch, WeMoLink, WeMoSwitch |
| 12 | Withings | Withings Wireless Scale WS-30 |

**Table 8 Manufacturer group classification results.**

| Approach | Branch | Device manufacturer | | Training time |
|---|---|---|---|---|
| | | Accuracy % | F1-score % | |
| MIND-CNN | – | 98.14 | 97.85 | 2 min 31 s |
| MIND-MLP | – | 96.09 | 95.9 | 2 min 10 s |
| Chowdhury et al. (2020) | (Ori) | 99.37 | – | – |
| | (Rep) | 98.58 | 93.02 | 4 min 2 s |

The MIND-IoT models demonstrate superior performance and robustness compared to traditional machine learning approaches, particularly when evaluated on different datasets. The pre-trained MIND-IoT models benefit from its transfer learning feature and generalize well across varied data, showcasing their potential for real-world IoT device classification tasks.

## Evaluation with manufacturer group classification

For additional MIND-IoT model benchmarking, this study includes an evaluation against the work of Chowdhury et al. (2020), which classified the IoT Sentinel dataset by device manufacturer groups, regardless of device type. There are a total of 12 groups of devices in the Sentinel dataset, as shown in Table 7.

The MIND-IoT model extends the fine-tuning task to include device manufacturer groups based on the IoT Sentinel dataset. This evaluation split the dataset into training and testing sets using a 70:30 ratio, with manufacturer groups serving as labels. The fine-tuning was conducted for only five epochs as the training losses stopped improving.

The results in Table 8 demonstrate the efficiency and effectiveness of the MIND-IoT models when fine-tuning for the new task of classifying devices by manufacturer group.

Both MIND-CNN and MIND-MLP achieved high accuracy and F1-scores with significantly less training time compared to the replicated results of *Chowdhury et al. (2020)*. Specifically, MIND-CNN achieved an accuracy of 98.14% and an F1-score of 97.85% within just 2 min and 31 s of training time, while MIND-MLP achieved 96.09% accuracy and a 95.90% F1-score in 2 min and 10 s.

In comparison, the replicated results of Chowdhury's approach achieved a slightly higher accuracy of 98.58% but a lower F1-score of 93.02%, with a longer training time of 4 min and 2 s. The original results reported by Chowdhury showed an accuracy of 99.37%, though the F1-score was not available.

These results highlight the MIND-IoT models' ability to quickly adapt to new tasks with minimal fine-tuning, showcasing the pre-trained model's robustness and generalizability. The efficiency in training time is a significant advantage, particularly in practical applications where rapid adaptation to new data is crucial. The seamless integration and fine-tuning of the MIND-IoT models for this new task demonstrate the effectiveness of the pre-trained model in fusing with related tasks, ensuring high performance and adaptability.

## CONCLUSIONS

This study comprehensively evaluated the MIND-IoT models for IoT traffic classification, emphasizing their performance across diverse datasets and classification tasks. The rapid expansion of the IoT has led to a significant surge in network traffic volume and diversity. Accurate IoT traffic classification is essential for maintaining network security, optimizing resource allocation, and ensuring efficient device management. However, current methods are often limited to localized, single-task applications and struggle with the complexity and diversity of modern IoT environments, lacking robustness and generalizability. Existing deep learning solutions generally face limitations in generalization across dynamic IoT networks and diverse device types, requiring extensive retraining or costly data annotation, which is further worsened by their dependency on labelled datasets. This issue highlights the urgent need for models that can efficiently learn and adapt to generalized IoT traffic classification.

To address these challenges, this study introduced MIND-IoT, a scalable framework that combines pre-trained Transformer-based architectures with CNN. Our objective was to develop a generalized solution for IoT traffic classification that enhances robustness, adaptability, and performance across diverse IoT environments and classification tasks. The model operates in two phases: a pre-training phase, where it utilizes MLM to learn from large-scale IoT data, and a fine-tuning phase that adapts the model to specific tasks, such as binary classification of IoT *vs.* non-IoT, IoT category classification, and device identification.

Our key achievements demonstrate MIND-IoT's superiority over traditional methods. We demonstrated that increasing model size generally reduces training losses, although performance gains diminish with excessive complexity, highlighting the importance of balanced configurations for practical deployment. The single-task MIND-IoT models, particularly MIND-CNN and MIND-MLP, consistently outperformed traditional machine

learning methods in binary IoT *vs.* non-IoT classification, IoT category classification, and IoT device type classification. Benchmarking against existing works further validated their superiority in handling diverse and dynamic IoT traffic scenarios. A significant advantage of MIND-IoT is its ability to adapt to new classification tasks rapidly with minimal fine-tuning, showcasing robust performance and generalizability, particularly notable in its efficiency in training time for real-world applications. This is attributed to the model's capacity to capture generalized and context-aware patterns, including relationships between packet-level and flow-level statistics, through the attention mechanism used during pre-training.

While MIND-IoT demonstrates strong performance, its high computational complexity and limited interpretability present challenges for practical deployment, especially in resource-constrained IoT environments. Future work should focus on optimizing the MIND-IoT model for computational efficiency through techniques such as model compression, knowledge distillation, or pruning. These improvements will make MIND-IoT more adaptable, efficient, and trustworthy for dynamic, real-world IoT deployments.

## ACKNOWLEDGEMENTS

Grammarly was used to enhance written content through proofreading, clarity suggestions, and tone refinement.

### Funding

This work was supported by Konsortium Kecemerlangan Penyelidikan (JPT(BKPI)1000/016/018/25 (49)) and the Fundamental Research Grant Scheme (FRGS/1/2023/ICT11/UM/02/1) provided by the Ministry of Higher Education of Malaysia. The funders had no role in study design, data collection and analysis, decision to publish, or preparation of the manuscript.

### Grant Disclosures

The following grant information was disclosed by the authors:
Konsortium Kecemerlangan Penyelidikan: JPT(BKPI)1000/016/018/25 (49).
Fundamental Research Grant Scheme: FRGS/1/2023/ICT11/UM/02/1.
Ministry of Higher Education of Malaysia.

### Competing Interests

The authors declare that they have no competing interests.

### Author Contributions

- Firdaus Afifi conceived and designed the experiments, performed the experiments, analyzed the data, performed the computation work, prepared figures and/or tables, authored or reviewed drafts of the article, and approved the final draft.

- Faiz Zaki analyzed the data, prepared figures and/or tables, authored or reviewed drafts of the article, and approved the final draft.
- Hazim Hanif conceived and designed the experiments, analyzed the data, performed the computation work, prepared figures and/or tables, and approved the final draft.
- Nik Aqil performed the experiments, analyzed the data, prepared figures and/or tables, and approved the final draft.
- Nor Badrul Anuar conceived and designed the experiments, performed the experiments, analyzed the data, authored or reviewed drafts of the article, and approved the final draft.

## Data Availability

Code is available at Zenodo:

fdaus. (2025). fdaus/MIND-IoT: First Mind-IoT Release (v1.0.0). Zenodo. https://doi.org/10.5281/zenodo.15725619.

The FCSIT dataset is available at Figshare: Aqil (2024). IoT-FSCIT. figshare. Dataset. https://doi.org/10.6084/m9.figshare.25143581.v1.

The IOT Traces dataset is available at the University of New South Wales: https://iotanalytics.unsw.edu.au/iottraces.html#bib18tmc.

The Moniotr dataset is available at Northeastern University: https://moniotrlab.ccis.neu.edu/imc19/.

The IoTFinder Data dataset is available at YourThings: https://yourthings.info/data/.

The Iot Sentinel dataset is available at Aalto University: https://research.aalto.fi/en/datasets/iot-devices-captures.

DOI: 10.24342/285a9b06-de31-4d8b-88e9-5bdba46cc161.

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
