# Peer review of "Transformer-based tokenization for IoT traffic classification across diverse network environments"

_PeerJ Computer Science, doi:10.7717/peerj-cs.3126_

## Round 0.1 · original submission · Major Revisions

Reviewers have identified a number of issues that should be addressed.

**Language Note:** The review process has identified that the English language must be improved. PeerJ can provide language editing services - please contact us at [email protected] for pricing (be sure to provide your manuscript number and title). Alternatively, you should make your own arrangements to improve the language quality and provide details in your response letter. – PeerJ Staff

Reviewer 1 ·

Basic reporting

Overall, the English language needs to be improved.
Your problem and objective need to be clearer and more concise in the introduction section.
The manuscript structure is good and sequenced.

Experimental design

The overall design is good and clear.

Validity of the findings

This research novelty is good, but you must justify your work with some of the latest research works.

The conclusion needs to be more concise and clear.

Additional comments

The study relies heavily on older references (2018–2022), with minimal inclusion of recent research (none from 2024 and only two from 2023). Include more recent works, especially from 2022–2024.
Revise the language thoroughly to improve the manuscript's quality.
Add a supporting reference for Line 169.
Ensure proper referencing for Lines 232, 236, and 242.
Provide a brief description of the IoT-FCSIT dataset, including how it was collected and details about the devices.
Include a sample of raw data and a list of extracted features.
Expand abbreviations (e.g., MAC, ARP, DHCP) before using them.
Ensure consistent font type, such as in Lines 384 and 385.
Use a lowercase letter in Line 395 where needed.
Use uppercase letters for “FFN” in Line 408, consistent with other terms.
Ensure consistency in formatting, such as in Lines 199 and 468.
Avoid repeating full forms, as seen in Line 470.
Include the missing category list.
Add equations for assessment metrics in the manuscript.
Correct “MIND-IoT” in Lines 559 and 673.
Review and compare your work with the latest relevant studies, such as:
Identifying SH-IoT devices from network traffic characteristics using random forest classifier
A Deep Learning Approach for Classifying Network Connected IoT Devices Using Communication Traffic Characteristics
Summarize the conclusion section with:
The problem addressed
The objective of the study
Key achievements
Future directions

Reviewer 2 ·

Basic reporting

All figures are blurry. Please redraw them and use high-resolution figures. The Introduction section needs to be elaborated. What is the inference out of the background studies? List the key objectives of the proposed work.

Experimental design

I could not find any experimental graphs. It would be better to have state-of-the-art analysis to prove the superiority of your work.

Validity of the findings

It would be better to illustrate the proposed approach with sample traffic data.

Reviewer 3 ·

Basic reporting

This paper proposed a method using Transformer based Pre-training and CNN based classifier for IoT traffic classification. It might be better to review the SOTA method with transformer based traffic classification, since there are many studies focusing on this.

Experimental design

The pre-training methods typically perform well than traditional training style since it will be fed a lot of data. As the datasets mentioned in this paper, it appears to be scarce.

Validity of the findings

As for the parser, it can not validate the syntactic and contextual relations only using a pair of feature names and value, because model can not read and learn the relationship between the features if you do not tell the model using other way. For example, how does the model can learn the relationship between the packet length and flow length?

Reviewer 5 ·

Basic reporting

The paper is okay overall. However, following comments need to be addressed.
• Is this the first study that uses a transformer-based model for traffic classification? I found some existing studies. Is this the first one in IoT? Not clear.
• Why was standard tokenization techniques not applicable to this use case? Why did you design your own tokenization technique?
• The authors have very well-reviewed on the background on traffic classification. However, the current definition of traffic classification is shallow. Need to elaborate on it mentioning what are the different types of data collected for traffic classification with respect to insight from recent studies (10.3390/telecom4030025) that have reviewed on traffic classification tasks. That is because different techniques can collect diverse data based on the application.
• Also mention which type of data you focus on and justification for why you select that data.
• What was the transformer model that you incorporated in your research. Couldn’t find it? Did you develop your own custom one? Is it a custom MLP? If yes, giving its architecture diagram will be better.
• Why was the particular data set selected? Did you train/fine tune the model using the same data set? Will it generalize over the other datasets? Add explanations
• In conclusion, the authors have very well explained the conclusions, limitations and future directions.
With these minor modifications, the paper can be significantly improved and can make a good contribution.

Experimental design

Experiments are okay. But, some explanations needed as per my previous comments.

Validity of the findings

Experiments are validated and benchmarked.

Additional comments

Improve the paper considering my comments.

·

Basic reporting

The paper proposes a Transformer model called MIND-IoT for IoT network traffic classification. An important feature of the model is that it is pretrained self-supervised with a simple masking approach, and only needs to fine-tune a classification head for specific tasks or datasets.
Strengths:
1) The paper is very well-written and easy to follow.
2) The motivation is convincing; the need for such a model is supported well by arguments such as the scarcity of labelled data
3) The literature review is good
4) The methods (transformer, new tokenisation scheme, fine-tuning head) are novel, valid and are motivated well.

My main concerns regard the experimental design (see below), here are some minor remarks:
- The paper is quite long. I would suggest to shorten a bit by removing repetitive parts (e.g. contributions are listed in different forms three times as a-d: in the intro, the background section and the beginning of the methods section) and by avoiding explanations of widely used methods, such as general metrics such as “Accuracy” (it’s an established metric, no need to explain them) or explaining a feed forward layer of a neural network.
- Line 359: wrong grammar in “as summarized in Table 2 summarises 64 pre-defined tokens used in this study”
- Line 545 “This study pre-trains three Base-IoT models with different hyperparameter configurations for more than 20,000 training steps” -> this sentence seems to repeat a previous sentence and contradict it (20k or 30k steps?)
- Line 709 error in “its performance drop on unseen datasets like Sentinel and Mind-IoT” should be Sentinel and the other dataset.
- Line 591 “300 thousand” rewrite to 300k or 300000
- Maybe rename "IoT-Base" to "IoT-Large", since the medium model is used for all the experiments, so “base” is a bit misleading.

Experimental design

The experimental design is generally good, since several public datasets are used and there is a comparison to baselines from the literature.
However, I have some major concerns and questions:
1) The split in training and testing data is not entirely clear. Isn’t the idea of the foundation model that you would pretrain on some datasets, and then fine-tune on other datasets for a specific task? Here, it seems like the models were pretrained and fine-tuned on the same datasets, at least for the baseline study it says “All classification models were trained and fine-tuned on the same feature set using the UNSW IoT Traces and MonIoTr datasets”. Could you specify for each experiment what data was used for pretraining and fine-tuning for your method, and what data was used to train the baseline methods (RF, CNN, MLP, etc)?
3) The data augmentation scheme is not clear to me. The method is “inspired by Chu & Lin (2023) work that combines a Multilayer Perceptron (MLP) with GANs” (line 295) - what does “inspired mean? Is the same method used or what is changed? And what data is used for training and testing the performance during data augmentation? When is this training performed - is it done before the pretraining phase or is it integrated into the pretraining or finetuning?
4) There are no ablation studies with respect to specific architectural choices, e.g. for showing whether the new tokeniser makes a significant difference.
5) It would be good to include baselines with the same approach, i.e. pretraining on one dataset and fine-tuning on another.

Validity of the findings

The results generally support the statements in the introduction and conclusion about the superiority of the new approach, but I am not entirely sure about their validity:
1) In all experiments, the MIND-IoT model performs quite well on Sentinel and FCSIT datasets, whereas the performance of other models drops by 50%. This looks a bit to me as if MIND-IoT was fine-tuned on these datasets, whereas the other models were not. If this is the case, the comparison would be highly unfair. Was this the case, or were all models trained just on the UNSW and MonIoTr datasets?
2) Sentinel dataset not shown in Table 6, but mentioned several times in line 621ff, where Table 6 is described and interpreted. Please make sure that the table is consistent with the text describing it.
3) I disagree with the conclusions drawn from Table 5. The table basically shows that pretraining on other datasets hardly improves the results, because training just on the UNSW dataset is pretty much the same performance than pretraining on all datasets and testing on the UNSW dataset.

---

## Round 0.2 · accepted · Accept

The reviewers are satisfied with the recent changes so I can recommend this article for publication.

Reviewer 3 ·

Basic reporting

-

Experimental design

-

Validity of the findings

-

·

Basic reporting

I appreciate the thorough revision, which addresses all my comments. I recommend acceptance of the paper.

Experimental design

The experimental design is now clarified and is suitable from my point of view.

Validity of the findings

The findings (eg, low performance of baselines on other datasets) are explained better in the revised manuscript. They are valid.